# Worldwide Selection Footprints for Drought and Heat in Bread Wheat (*Triticum aestivum* L.)

**DOI:** 10.3390/plants11172289

**Published:** 2022-09-01

**Authors:** Ana L. Gómez-Espejo, Carolina Paola Sansaloni, Juan Burgueño, Fernando H. Toledo, Adalberto Benavides-Mendoza, M. Humberto Reyes-Valdés

**Affiliations:** 1Programa de Doctorado en Recursos Fitogenéticos para Zonas Áridas, Universidad Autónoma Agraria Antonio Narro (UAAAN), Saltillo 25315, Mexico or; 2International Maize and Wheat Improvement Center (CIMMYT), Texcoco 56237, Mexico

**Keywords:** *Triticum aestivum* L., landraces, adaptation, drought stress, heat stress, Genome–environment Associations (GEA)

## Abstract

Genome–environment Associations (GEA) or Environmental Genome-Wide Association scans (EnvGWAS) have been poorly applied for studying the genomics of adaptive traits in bread wheat landraces (*Triticum aestivum* L.). We analyzed 990 landraces and seven climatic variables (mean temperature, maximum temperature, precipitation, precipitation seasonality, heat index of mean temperature, heat index of maximum temperature, and drought index) in GEA using the FarmCPU approach with GAPIT. Historical temperature and precipitation values were obtained as monthly averages from 1970 to 2000. Based on 26,064 high-quality SNP loci, landraces were classified into ten subpopulations exhibiting high genetic differentiation. The GEA identified 59 SNPs and nearly 89 protein-encoding genes involved in the response processes to abiotic stress. Genes related to biosynthesis and signaling are mainly mediated by *auxins*, *abscisic acid* (*ABA*), *ethylene* (*ET*)*, salicylic acid* (*SA*), and *jasmonates* (*JA*), which are known to operate together in modulation responses to heat stress and drought in plants. In addition, we identified some proteins associated with the response and tolerance to stress by high temperatures, water deficit, and cell wall functions. The results provide candidate regions for selection aimed to improve drought and heat tolerance in bread wheat and provide insights into the genetic mechanisms involved in adaptation to extreme environments.

## 1. Introduction

Since its domestication more than 10,000 years ago, common wheat has experienced a series of selective events caused by humans and the environment, contributing to the increase in its genetic diversification [1]. Climate change has severely reduced wheat production in recent years due to extreme temperature episodes and unpredictable precipitation patterns [2,3]. Simulation models predict losses of more than 20% in agricultural production by 2050 [4].

There is an urgent need to discover new sources of adaptation to drought and heat that contribute to maintaining crop productivity [2]. To address this scenario, landraces are valuable, because they have developed survival mechanisms for challenging environments through natural and human selection [5,6]. This is why they preserve loci of adaptation to climate change in their places of origin [7].

Recent advances in sequencing technologies have allowed the exploration of entire genomes in various species with increasingly dense *single-nucleotide polymorphism* (SNP) data that identify selective events [8]. Likewise, novel bioinformatic analysis approaches, such as genome-wide association studies (GWAS), are very efficient in time, cost, and precision for identifying genes that control important agricultural traits [9].

Genome–environment Associations (GEA) or Environmental Genome-Wide Association scans (EnvGWAS) have been used successfully for studying adaptive traits in local populations. They consist of associating SNPs distributed throughout the genome with environmental variables of the accession sampling sites [10].

Associations between the genome and environment of origin were initially documented in wild populations, with the successful identification of adaptive loci and prediction of phenotypic variations [11,12,13,14]. However, its application in crops is recent, with research carried out on sorghum through bioclimatic and soil gradients to predict adaptive traits [15]. Subsequent applications have been made in crops such as corn [16], beans [10,17], barley [7,18], soybean [19], tomato [20], chickpea [21], peach [22] and wheat [23].

In the last decade, significant progress has been reported in the characterization of wheat genomes through high-throughput genotyping with DArT-seq technology. More than 100,000 accessions belonging to the germplasm bank of the International Maize and Wheat Improvement Center (CIMMYT) have been characterized through the Seeds of Discovery initiative [24,25].

Considering the effectiveness of GWAS for the identification of genomic regions associated with traits of agronomic importance, as well as the functional genetic variation to adapt crops to climate change, this research aims to identify genomic regions related to the adaptation process to arid climates through Genome–environment Association studies in the *Triticum aestivum* collection maintained in the CIMMYT germplasm bank.

## 2. Results

### 2.1. Exploratory Analysis

As expected, the highly significantly (*p* ≤ 0.001) and correlated variables share with each other temperature in their definition (Table 1). For instance (Table 1a), AMT was consistently significantly and positively correlated with the variables MaxTWM (r = 0.79), MeanTDQ (r = 0.71), and MeanTWQ (r = 0.77), similar to MaxTWM with MeanTWQ (r = 0.94). For those based on precipitation, only PDM with PDQ (r = 0.99) exhibited a significant correlation. Similarly, Table 1b shows positive correlations between the variables associated with temperature, such as the case of AMT with the variables MaxT (r = 0.98), HITmead (r = 0.92), and HITmax (r = 0.93); MaxT with HITmead (r = 0.92) and HITmax (r = 0.96); and HITmead with HITmax (r = 0.97). On the other hand, negative correlations were observed only for MeanTDQ with PWQ (r = −0.84) and AP with DI (r = −0.83), which reflect contrasting trends between temperature and precipitation.

In the PCA, we observed that the evaluated variables had contrasting contributions to the total variation of each component, especially for the second set of variables (Appendix A). In the biplot graphs of the PCA (Figure 1), we identified that the accessions sites are well-differentiated with respect to their Köppen-Geiger climate, revealing a greater representation of collections from the temperate (C) and cold (D) groups. For the first set (Figure 1a), those related to temperature (AMT, MaxTWM, MeanTDQ, and MeanTWQ) are oriented along PC1, while the precipitation-derived variables (PDM, PS, and PDQ) predominate in PC2. Similarly, for the second set (Figure 1b), PC1 includes the temperature-related variables (AMT, HITmead, MaxT, and HITmax), while PC2 includes AP and DI. The PCA helpfully identified some variables with discriminating potential of accessions according to the Köppen-Geiger climate groups (Figure 1b). For example, the high DI sites are related to dry climates (B), whereas the temperature-related variables (HITmax, HITmead, AMT, and MaxT) define gradients from tropical (A) to temperate (C) groups. On the other hand, we identify the variables of most significant importance concerning the monitoring, follow-up, and informativeness of drought and heat stress events and their contribution to the total variation.

### 2.2. Population Structure

The cross-entropy validation implemented in the LEA package, based on SNP markers, suggested an optimal number of 10 subpopulations (Figure 2). This subdivision reflects diversity according to the climate in the subpopulations, since there is no explicit representation between these and their geographic or regional origin. The main contribution of accessions comes from Turkey and China of 49.5% and 27.8%, respectively. This is followed by Afghanistan, Tajikistan, and Iran of 5.2%, 4.2%, and 3.4%, respectively. The rest of the countries contribute with less than 2% of the accessions. Subpopulations II, IV, VI, and X constitute 60.4% of the total population under study. On the other hand, the dispersion pattern observed in the molecular PCA biplot revealed that the subpopulations were well-differentiated, reflecting the high genetic diversity of the analyzed accessions (Figure 3a). The correspondence analysis (Figure 3b) between the subpopulations and Köppen-Geiger climate groups showed that dry weather accessions (B) were better associated with subpopulation VII. This subpopulation comprised 83 accessions from eight countries (Afghanistan, Armenia, Azerbaijan, China, Iran, Iraq, Tajikistan, and Turkey) cataloged principally within the region of Southern Asia (SAS) with records of an aridity index (DI) greater than 5, whose values indicated water deficiency.

### 2.3. Genome-Wide Association Studies

We identified 59 SNP markers associated with the climatic variables evaluated in all 21 bread wheat chromosomes (Table 2). The chromosomes with the highest number of associated markers were 2B (seven SNPs), 7A (six SNPs), 3B (five SNPs), and 5B (five SNPs). The chromosomes with a single associated marker were 1B, 5A, 6A, 6D, and 7D.

For all variables, the QQ plots (Figure 4 and Figure 5) show a good adjustment, with most −log_10_ (*p*-values) for the null hypothesis of no association, being close to the diagonal. In contrast, some points at the top of each plot may be in *linkage disequilibrium* (LD) with a causal polymorphism, indicating that the model has a reasonable control for both false positives and negatives.

Regarding the individual detection of association for each variable, there were different levels of associated SNPs (Appendix A). These differences in detecting different SNPs in highly correlated variables may be due to the possible presence of atypical data in the collection sites. Likewise, it is known that the effects produced by drought and heat are differential in certain phases of reproductive development, during which plants are more susceptible. We observed a lower association with variables DI, AMT, and AP, with less than eight markers for each one. In contrast, the variables with more associated loci were PS, HITmead, MaxT, and HITmax, with 15, 14, 12, and 11 SNPs, respectively.

At least 10 SNP located in seven chromosomes (1A, 3A, 3D, 4A, 4D, 5B, and 6D) were detected more than once for variables AMT, MaxT, HITmead, and HITmax. The most frequent SNPs were 108825112|F|0-19:T>A-19:T>A and 108891114|F|0-33:A>G-33:A>G on chromosome 1A and allele 109431634|F|0-54:G>A-54:G>A on chromosome 4D. These SNPs are related to the genes *Auxin response factor* (*ARF*), *CONSTANS-like* (*COL*), and *proteins abundant during late embryogenesis* (*LEA*).

### 2.4. Identification of Genes Related to Adaptation to Abiotic Stress in Plants

For the associated regions, we found 89 candidate genes encoding proteins related to various biological processes in plants (Table 2). Among these, we identified the significant presence of 26 proteins involved in the signaling network (Table 3), 15 cell wall structural proteins (Table 4), 21 response proteins to various types of abiotic stress (Table 5), and 7 proteins related to morphological changes (Appendix A).

Given that a response to stress begins with the perception and signal transduction of environmental stimuli, it is not surprising that we found an abundance of signaling proteins with a well-documented role in plant responses to drought and heat stress. This is the case for protein *kinases serine-threonine* (*Pstp*-2A; *STPK*-2B, 3A, 4A, and 4D; and *CIPK2*-2B) and some proteins activated by stress-related plant hormones, such as *ethylene* (*ET*), *abscisic acid* (*ABA*), *jasmonic acid* (*JA*), *salicylic acid* (*SA*), and auxins, reported mainly by the BIO15, HITmax, and MaxT variables and chromosomes 7A, 4D, 2B, and 1D (Table 3).

Within the genes associated with the cell wall (Table 4), proteins involved in the biosynthesis of cuticle and cell wall components stand out, such as cuticular wax (*ASAT1*, *CCR*, and *KCS*); polysaccharides such as *cellulose*, *hemicellulose*, and *pectin* (*STL2* and *MAN2*); *lignin* (*OMT*); and *glycoproteins* (*GT* and *1,3-β-glucanase*) on chromosomes 2B, 2D, 3A, 3D, 5B, 5D, and 7A. We also found some structural proteins (*WAK90*, *EXPB2*, *RD22*, and *GT2*); a membrane trafficking protein (*ArfGAP*); and two signal transduction proteins (*MARCH* and *QKY*) on chromosomes 1A, 1D, 2B, 3D, 5B, and 7B.

Many morphological changes are induced when the plant is subjected to long periods of environmental stress (Appendix A). For this reason, we also observed some genes regulating multiple development processes (*FBX*-2B and 4D), organ morphology (*OFP*-2B), stomatal differentiation (*SCRM2*-4B), cell expansion (*GIF1*-4B), proteins that affect gravitropism (*AGD12*–6B), and sheet curling (*ROC5*-6B).

### 2.5. Drought and Heat Adaptation Genes

Through genomic association with the variables AP, PS, MaxT, HITmead, and DI, we identified 12 genes related to the response and tolerance of plants to drought and heat stress (Table 5 and Figure 6) on chromosomes 1D (one gene), 2B (two genes), 4D (one gene), 5A (one gene), 5B (three genes), 5D (one gene), 6A (one gene), and 7A (two genes). Proteins from different domains represent biological processes related to water deficit: enzymes with *galactinol synthase* activity (*GolS*-2B); *leucine-rich repeat kinase proteins* (*LRRK*-2B, 6A, and 7A); *late embryogenesis abundant proteins* (*LEA*-4D); *BURP* domain proteins (*BURP*-5B); *WRKY* transcription factors (*WRKY*-5B); and component box proteins of the *SKP–Cullin–F-box E3 complex* (*SCF*-7A). Likewise, the response to heat stress by some *heat shock proteins* (*HSP20*-1D and *HSFA2E*-5A), a *chronic heat stress thermotolerance protein* (*GrpE*-5B), and a *DNA-dependent RNA polymerase* (*NRPB1*-5D).

### 2.6. Other Genes

Additionally, we observed seven genes involved in the response to biotic stress (*PELO*, *RIP*, *NBS-LRR*, *LFG4*, *FMO*, and *CNGC2*); six genes involved in photosynthesis (*GLO1*, *CHLH*, *PSB33*, *psbL*, *γCA1*, and *PETG*); four genes involved in flowering (*COL*, *CK2*, *FPF1*, and *APK*); and three genes involved in nutrient assimilation (*YSL1*, *PAP*, and *wSs2a-3*) on chromosomes 1A, 2A, 2B, 2D, 3A, 3B, 3D, 4A, 4D, 5B, 5D, 7A, 7B, and 7D (Appendix A).

## 3. Discussion

### 3.1. Environmental Variables Involved in the Detection of Adaptive Loci by GEA

Climate change affects diverse geographic areas throughout the world. However, its effects in arid and semiarid climatic zones have devastating impacts [26]. These selective environmental effects play an essential role in the local adaptation, genetic diversity, and population structure of wild accessions [17]. Therefore, using climatic variables to represent selective environmental pressure can be valuable to capture important components of the mechanisms of resistance and tolerance to abiotic stress. We observed local adaptation footprints in multiple genomic regions along the 21 wheat chromosomes, with each climatic variable having different numbers of genomic associations of biological importance.

It should be remarked that the climate data came from records spanning 1970–2000, which naturally did not cover the adaptation period of the accessions. However, these 30-year records are good indicators of the prevailing climate type in the collection sites. On the other hand, the accessions were collected from 1983 to 2011, with the main bulk of the collection occurring in 1984 and 2011. One must be aware of the noise arising from the migration dynamics of those materials, especially for the most recent collection efforts. Thus, the herein reported results are subject to validation by different approaches.

Exceptionally, the seasonality of precipitation (PS) had the largest number of significant SNP markers along different chromosomes, placed close to unique genes of resistance and tolerance to abiotic stress. This makes sense, because it is considered an important variable in influencing the distribution of species through water availability [27]. On the other hand, measuring the variations in precipitation [28] at the sites of origin of the collections over three decades (1970–2000) faithfully represents the alterations in the uniformity and distribution of precipitation.

The research of Cortés et al. [29] reaffirmed the above; they found a strong influence of rainfall patterns on the population structure and the ecological diversity to tolerance drought in wild beans. Usually, prolonged periods of drought cause the expression of genomic regions associated with the activation of plant survival mechanisms [17]. 

The associations with the maximum temperature (MaxT), heat index (HITmead), annual precipitation (AP), and drought index (DI) identified several adaptation genes to drought and heat stress. This is primarily explained by the nature and importance of these variables in the monitoring of conditions of meteorological drought. Both AP and DI are valid descriptors for measuring the drought intensity [30]. DI is calculated through a combination of climatic and meteorological variables, among which precipitation is the most important [31]. In addition, the estimate values DI presented an excellent discriminating potential of accessions from arid climates (B), which gives reliability to its use. On the other hand, according to López-Hernández & Cortés [10], the maximum temperature and the heat index are better estimators of the natural adaptation to high temperatures and identify successfully associated genetic factors markers.

Different variables shared associations with some loci, suggesting that their selective pressures can shape the same genomic regions [22] and, therefore, remain stable in the landraces of *Triticum aestivum*. The most frequent loci on chromosomes 1A and 4D are related to two genes: *CONSTANS-like* and *proteins abundant in late embryogenesis* (*LEA*). The first is involved in various biological processes of plants, such as the control of the flowering time, regulation of growth and development, and responses to abiotic stress [32,33,34]. *LEA* proteins are recognized during the adaptation to abiotic stress, which includes dehydration, salinity, high temperature, and cold [35,36,37].

On the other hand, the only matching locus between AP and DI flanked an F-box domain gene, which is a homolog of the *GrpE* protein. In Arabidopsis, this gene acts as a nucleotide exchange factor of the 70-kD *heat shock protein complex* (*HSP70*), which specializes in thermotolerance to heat stress [38,39].

### 3.2. Adaptation to Drought and Heat Stress

Despite their coexistence in a climate change scenario, the combined effects of drought and heat stress have been poorly studied [40]. They have a synergistic effect, altering the metabolism and gene expression in ways other than those induced independently [41]. These combined effects affect several physiological, cellular, and molecular processes in plant cells [10].

The stress response mechanism in plants is very complex and requires several integrated pathways to be activated in response to external stress [42]. Plant hormones, such as *auxins*, *abscisic acid* (*ABA*), *ethylene* (*ET*), *salicylic acid* (*SA*), and *jasmonates* (*JA*), operate together in the modulation of the plants’ heat and drought stress responses [43,44].

Typically, *auxin* and the *auxin* pathway regulate thermomorphogenesis in plants, coordinating the growth and defense against heat stress [45], while *ABA* and *ET* interact positively to activate or repress the expression of numerous stress response genes, such as *LEA* proteins and *dehydrins* [46,47]. Likewise, *SA* is related to the synthesizing of protein chaperones, heat shock proteins, protective membrane proteins, antioxidants, and secondary metabolites [46,48,49]. *Jasmonates* significantly improve the tolerance to heat and drought stress through various *TFs*, which induce responsive gene expression and organic osmo-protectant activation, osmotic adjustment, and antioxidant activity [50,51,52].

Many biochemical and physiological impacts affect the growth and development of plants [40], as they affect the photosynthetic system, gas exchange, and water relations [45]. Consequently, a series of physiological and molecular responses are produced, which include root increase, reduction in the number and conductance of the stomata, decrease in leaf area, and morphological changes in leaves [53]. On the other hand, among the molecular responses, one should consider the production of antioxidants and osmolytes for osmotic adjustment and the expression of various proteins, such as *HSP*, *WRKY*, *MYB*, *LEA*, and *GrpE* [40].

The genes reported in this work are involved in most of the mentioned biological processes, including genes with signaling roles and genes associated with the cell walls and membranes, photosynthesis, flowering, and, of course, proteins involved in the response to heat and drought stress on various chromosomes. Our findings are consistent with Y. Li et al. [22]: plant genomes have been shaped by natural selection during local adaptation to different environmental conditions, so there is a close relationship between species survival and response to climate change.

## 4. Materials and Methods

### 4.1. Geographical Data

Through code written in the R language v.3.4.4 [54], the passport data of 174,553 accessions from the CIMMYT Wheat Germplasm Bank were filtered to select 1151 landraces of *Triticum aestivum* with unique and sensible geographic coordinates. Subsequently, the location mapping was carried out through the geographic information system QGIS version 2.18 [55]. This filtering process yielded 990 landraces with validated geographic data. The accessions came from 33 countries distributed in 13 geographic regions (Table 6) [56].

### 4.2. Genotypic Data

Germplasm genotyping was carried out through DArT-seq technology in CIMMYT under the Seeds of Discovery initiative for 45,871 accessions belonging to the wheat germplasm bank. The information was integrated using R v.3.4.4 [54] in a data table with the HapMap format containing the information of 86,683 SNP loci. The markers’ physical locations were obtained by reference genome sequences provided by Diversity Arrays Technology (wheat_ChineseSpring04), and only markers unambiguously located in the wheat genome were retained. Subsequently, they were filtered for quality control through a selection of the cleanest and most informative SNPs, with a maximum missing data rate of 20%, Shannon entropy greater than zero, and variants with a minor allele frequency (MAF) ≥ 2%. The filtered table contained 26,064 SNP loci in 990 landraces with geographic data.

### 4.3. Climatic Data

For each collection site, we extracted the values of altitude, temperature, precipitation, and eight bioclimatic variables related to drought and heat stress (Table 7) at a spatial resolution of 2.5 min (4.5 km) from the WorldClim platform (https://www.worldclim.org/ accessed on 31 July 2022) using the *getData* function of the R raster package version 3.5–15 [57]. Bioclimatic variables are derived from the monthly temperature and rainfall historical climate data from 1970 to 2000 in order to generate more biologically meaningful variables [58].

Additionally, we determined the Köppen-Geiger main climate groups (Table 8) through the R kgc package version 1.0.0.2 [59]. Köppen classification was constructed based on five vegetation groups that distinguish between plants of the equatorial zone (A), the arid zone (B), the warm temperate zone (C), the snow zone (D), and the polar zone (E); in the subclassification, the second letter considers the precipitation while the third letter the air temperature [60].

### 4.4. Index Estimation

Two heat indices (HIT) were estimated by the Thornthwaite model [61] using values of the mean and maximum temperature from 1970 to 2000. We denominated the HITmean and HITmax, respectively:(1)$$\;\;\;\;\;HIT_{mean}=\overset{12}{\underset{i=1}{\sum }}\left(tmean_{i}/5\right)ˆ1.514$$
(2)$$HIT_{max}=\overset{12}{\underset{i=1}{\sum }}\left(tmax_{i}/5\right)ˆ1.514,$$

For *tmean_i_, tmax_i_* > 0, where *tmean_i_* is the average monthly temperature, and *tmax_i_* is the maximum monthly, respectively, for the *i*th month.

Furthermore, a drought index was calculated [29]. This index is based on the relationship between the potential evapotranspiration and the annual precipitation of each collection site:*DI* = 100 × [(*PET* − *AP*) /*PET*],(3)
where *DI* is the drought index, *PET* is the potential evapotranspiration, and *AP* is the annual precipitation. In this index, negative values indicate excessive precipitation, while positive values indicate water deficiency. The calculation of *PET* was done with the *thornthwaite* function of the R SPEI package version 1.7 [62], through the values of the monthly average temperatures from 1970 to 2000 (*tmed_i_ > 0*) with the estimated solar radiation being based on the latitude of each collection site.

### 4.5. Exploratory Analysis of Climatic Variables

To look for patterns of relationships among climate variables, we used the Pearson correlation coefficient (r) and principal component analysis (PCA). The standardized variables are used in PCA to estimate the correlation matrix and determine the principal components (PC). The bioclimatic variables were grouped into two sets, with the first one containing bioclimatic variables related to aridity (AMT, TS, MaxTWM, MeanTDQ, MeanTWQ, AP, PS, PDM, PDQ, and PWQ). The second set included AMT; MaxT; AP; constructed indices (HITmean, HITmax, and DI); and ELEV. AMT and AP were included in both sets, because they are considered the main variables for indices related to aridity. The biplot graphs were constructed with the factoextra R package version 1.0.7 [63]. In the latter, the vector’s length and the angle’s cosine were used to group the variables into different groups. The Köppen-Geiger climate groups of each collection were also included.

### 4.6. Population Structure

The stratification of the collection was explored by two methods. The first one was based on the Landscape and Ecological Associations studies (LEA) package [64], with the SNPs coded in numerical form (0, 1, and 2). The *smnf* function was used to estimate the ancestry coefficients (K) with cross-entropy. This algorithm was executed with 10 replications and a K-value from 1 to 10. The optimal number of K was defined to assign genotypes to subpopulations according to estimates of individual admixture coefficients from the genotypic matrix. Subsequently, we visualized principal components through GAPIT R package version 3.0 [65]. We compared the results of the population stratification in subpopulations with the Köppen-Geiger climate groups using a correspondence analysis (CA), which reveals the close relationships between and within two groups of categorical variables based on data provided in a contingency table.

### 4.7. Association Analysis

We used 990 landraces of *Triticum aestivum* genotyped with 26,064 SNP loci and seven variables (AMT, MaxT, AP, PS, HITmead, HITmax, and DI) to run Genome–environment Association (GEA) studies with the Fixed and random model Circulating Probability Unification multiple-locus model “FarmCPU” [66] implemented in the R GAPIT (Genome Association and Prediction Integrated Tool) package version 3.0 [65]. FarmCPU is characterized by iteratively using two models, a linear mixed model (MLM) and a fixed-effects model, to select a set of markers associated with a trait of interest. 

The significant SNPs were determined according to the Bonferroni threshold to an alpha of 0.05, with a threshold value of -log_10_ (0.05/26,064) = 5.72, coupled by the visual interpretation of the Q-Q plots. The Manhattan and Q-Q plots were built with the R CMplot package version 4.0.0 [67].

### 4.8. Candidate Genes and Their Annotation

The sequence of the significant SNP markers was blasted in the wheat reference genome IWGSC_refseqv1.0 [68] published in the Ensembl Plants database (www.https://plants.ensembl.org/ accessed on 31 July 2022) to identify the candidate genes. For this, the genes found in the overlapping region and within one Mb upstream and downstream of the matched regions were selected as the candidate genes, and their molecular functions were determined. We identified the proteins in the UniProt (https://www.uniprot.org/ accessed on 31 July 2022) and InterPro (https://www.ebi.ac.uk/interpro accessed on 31 July 2022) databases encoded by the candidate genes, the functionality of the sequences, their domains, and classification. Finally, we elaborated a schematic representation of the physical map of bread wheat with the significant SNPs associated with response proteins to water stress and heat stress with the R LinkageMapView package version 2.1.2 [69].

## 5. Conclusions and Practical Implications

The results suggest that local adaptations have footprints along the 21 wheat chromosomes in multiple genomic regions. We found vital genes that include several critical points on the abiotic stress response mechanisms—highlighting a considerable number of signaling genes mediated by plant hormones, regulatory processes of the cell wall, morphophysiological changes, photosynthesis, flowering, and some response mechanisms to abiotic stress. 

We showed that the climatic variables estimated with historical data help capturing the environmental variability that occurred in the collection sites of the landraces. The variables of the maximum temperature, annual precipitation, precipitation seasonality, and heat and drought indices relevantly participated in identifying genes related to the response to water deficit and high temperatures. This confirms its representativeness in determining aridity in some climatic regions. 

This study points to 89 genes involved in the adaptation of bread wheat to its native habitats by association with seven specific climatic variables. The results are consistent with the idea that environmental pressure has modeled, through natural selection, the structure of genomic regions in local wheat populations over time. 

Our findings constitute a new resource to select accessions carrying alleles linked to specific climatic responses, which can be exploited through genomic prediction tools to select germplasms with genetic potential for adaptation to climate change.

## Figures and Tables

**Figure 1 plants-11-02289-f001:**
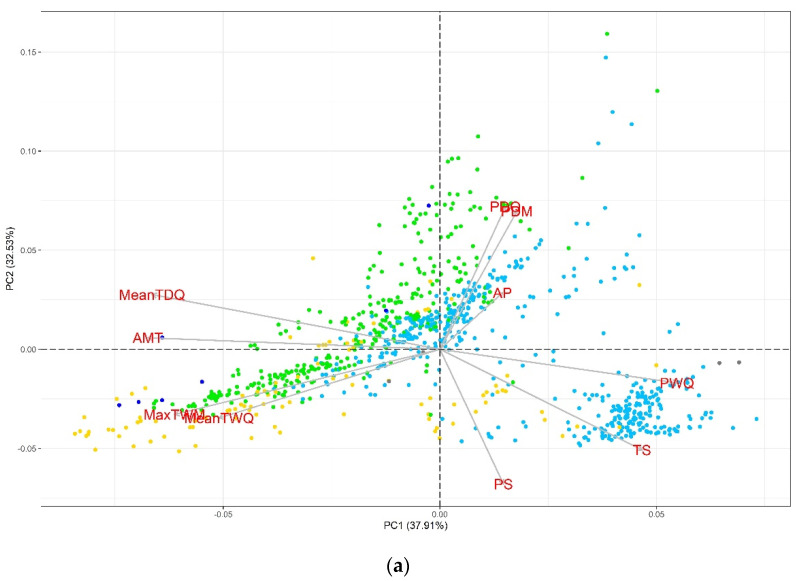
Principal component analysis for bioclimatic variables and elevation: (**a**) bioclimatic variables related to aridity and (**b**) temperature-related variables, calculated indices, AP, and ELEV. A = Tropical, B = Dry, C = Temperate, D = Continental, and E = Polar.

**Figure 2 plants-11-02289-f002:**
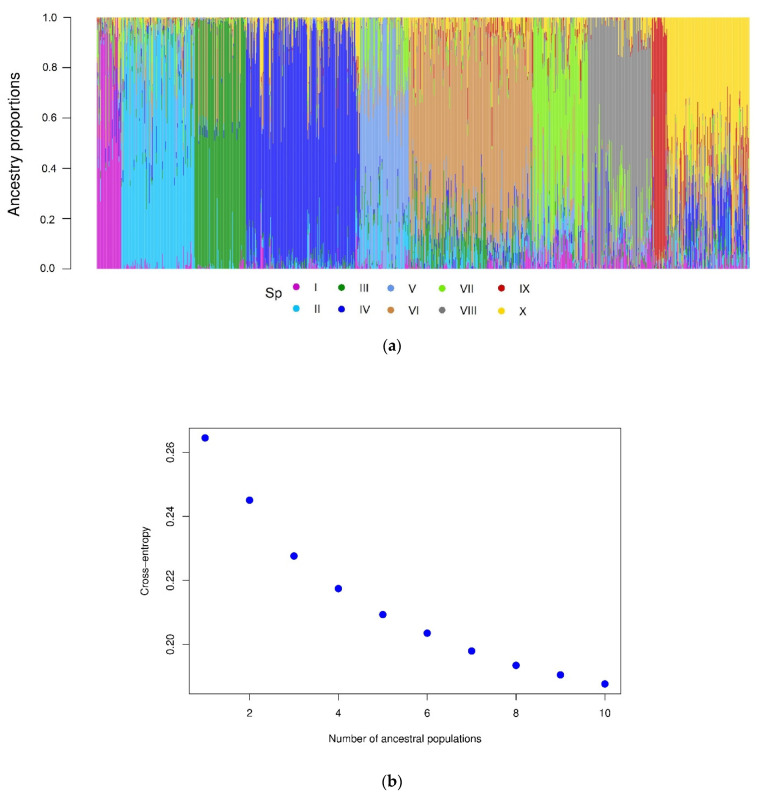
(**a**) Grouping of subpopulations (K= 10); each collection is represented by a thin vertical line, divided into colored segments representing the estimated probabilities of belonging (Q) to each subpopulation. (**b**) Ancestry coefficients estimated by LEA.

**Figure 3 plants-11-02289-f003:**
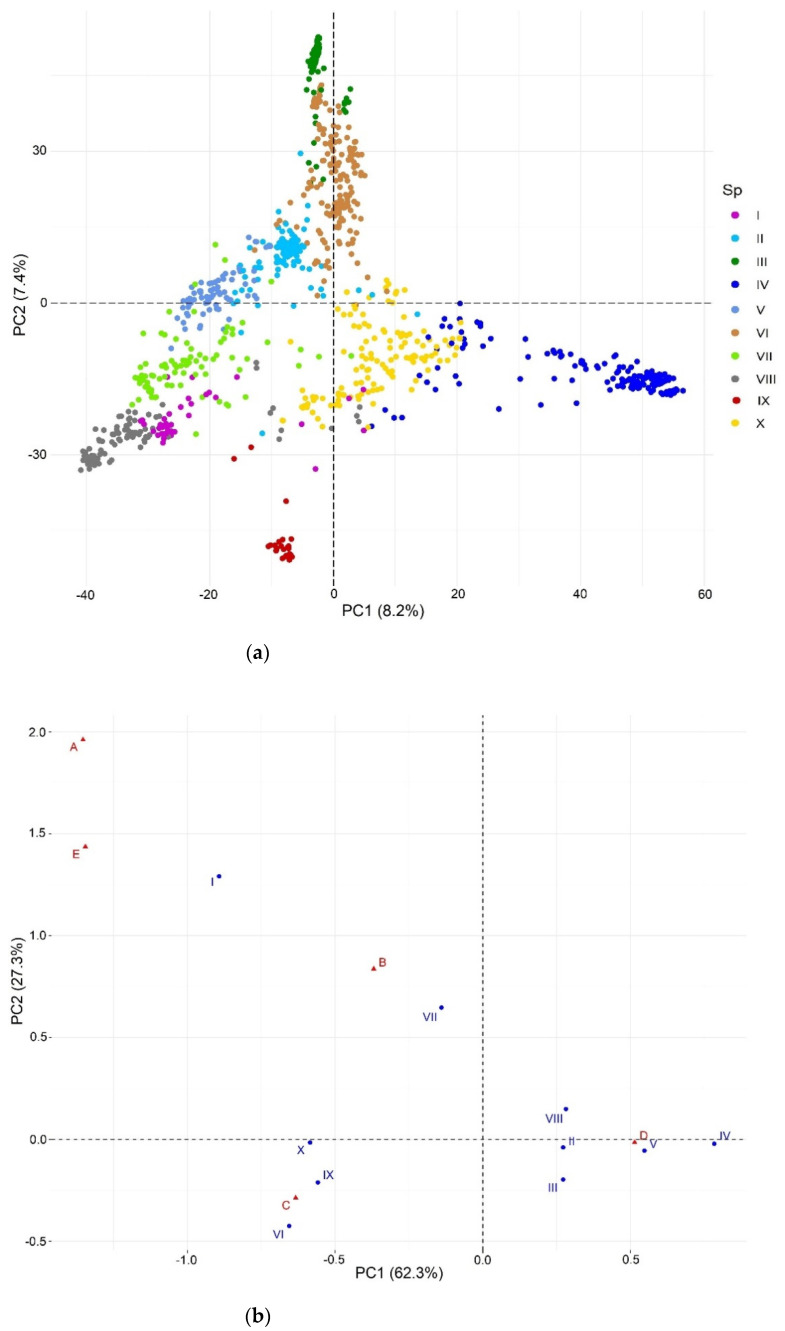
Population structure: (**a**) biplot for molecular PCA and (**b**) biplot of the correspondence analysis, where A = Tropical, B = Dry, C = Temperate, D = Continental, and E = Polar.

**Figure 4 plants-11-02289-f004:**
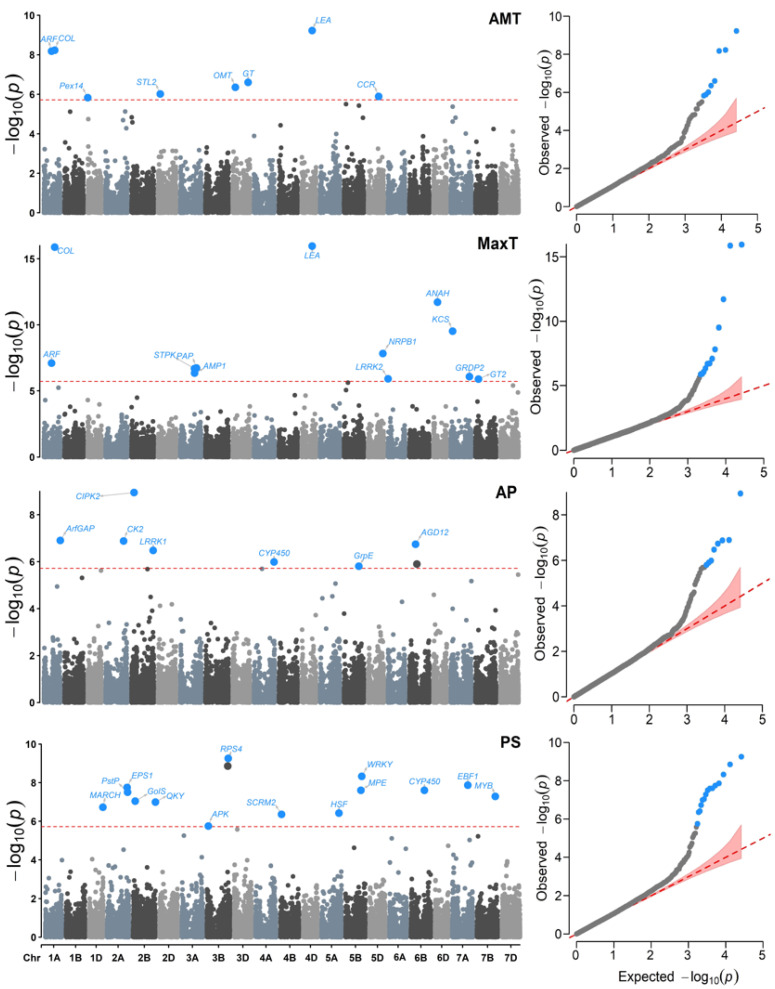
Manhattan (**left**) and QQ (**right**) plots for the Genome–environment Associations (GEA) analyses with the FarmCPU model and the variables: annual mean temperature (AMT), maximum temperature (MaxT), annual precipitation (AP), and precipitation seasonality (PS) in 990 landraces of bread wheat and 26,064 SNP markers. The red dashed horizontal line marks the -log_10_ (*p*-value) threshold after Bonferroni correction for multiple comparisons. The proteins associated with the significant SNPs are labeled in each graph.

**Figure 5 plants-11-02289-f005:**
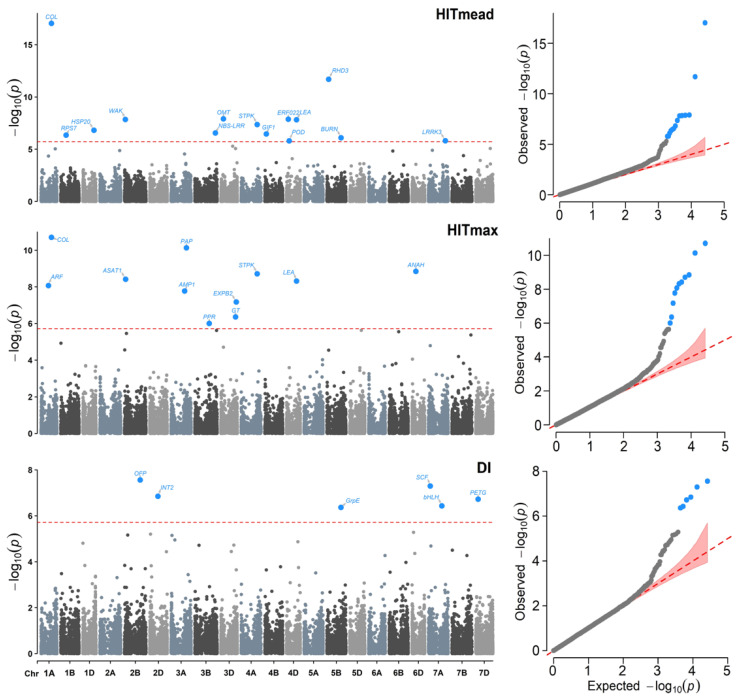
Manhattan (**left**) and QQ (**right**) plots for Genome–environment Association (GEA) analyses with the FarmCPU model and the variables: heat index of the mean temperature (HITmead), heat index of the maximum temperature (HITmax), and drought index (DI) in 990 landraces of bread wheat and 26,064 SNPs markers. The red dashed horizontal line marks the -log_10_ (*p*-value) threshold after Bonferroni correction for multiple comparisons. The proteins associated with the significant SNPs are labeled in each graph.

**Figure 6 plants-11-02289-f006:**
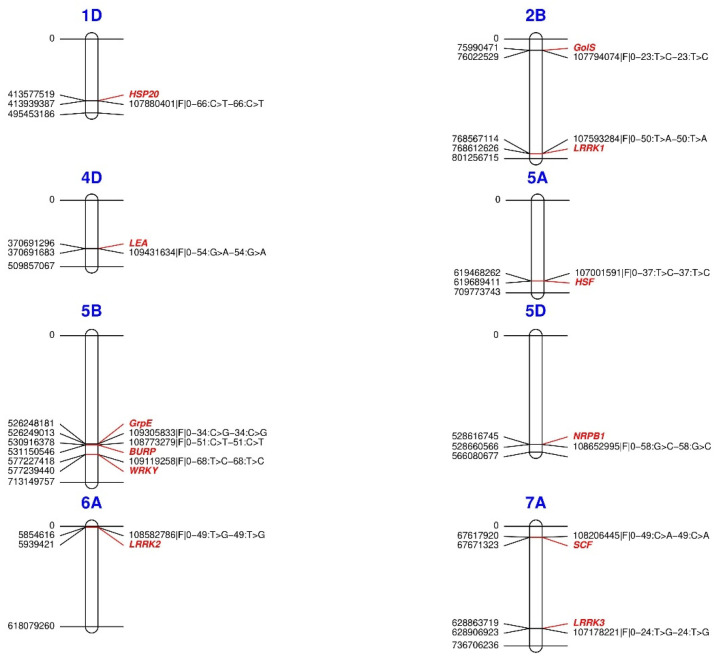
Physical map of the location of SNPs associated with heat and drought tolerance-related genes in *Triticum aestivum*. For each locus, the position is written on the left side, whereas SNPs are coded on the right side. Gene names are colored. *HSP20* = *Heat shock protein* class VI; *GolS* = *Galactinol synthase* 7; *LEA* = *Late embryogenesis abundant*; *HSF* = *Heat shock factor*; *GrpE* = *GrpE protein homolog*; *BURP* = *BURP* domain; *WRKY* = Probable *WRKY* TF 57; *NRPB1* = *DNA-directed RNA polymerase* subunit; *SCF* = *F-box component of the SKP-Cullin-F-box E3*; *LRRK* = *Leucine-rich repeat receptor-like kinase*.

**Table 1 plants-11-02289-t001:** Above the diagonal, correlations between bioclimatic variables, indices, and elevation. Below the diagonals, *p*-values for the significance test.

**(a)** **Correlation table for the bioclimatic variables related to aridity.**										
**Variable**	**AMT**	**TS**	**MaxTWM**	**MeanTDQ**	**MeanTWQ**	**AP**	**PDM**	**PS**	**PDQ**	**PWQ**
AMT		−0.65	0.79	0.71	0.77	0.07	−0.03	−0.08	0.01	−0.4
TS	0.000		−0.14	−0.76	−0.03	−0.13	−0.3	0.53	−0.35	0.56
MaxTWM	0.000	0.000		0.5	0.94	−0.17	−0.38	0.21	−0.38	−0.34
MeanTDQ	0.000	0.000	0.000		0.32	−0.08	−0.01	−0.5	0.03	−0.84
MeanTWQ	0.000	0.374	0.000	0.000		−0.02	−0.29	0.33	−0.29	−0.09
AP	0.023	0.000	0.000	0.009	0.552		0.41	0.01	0.44	0.38
PDM	0.408	0.000	0.000	0.645	0.000	0.000		−0.59	0.99	0.19
PS	0.012	0.000	0.000	0.000	0.000	0.814	0.000		−0.62	0.49
PDQ	0.784	0.000	0.000	0.357	0.000	0.000	0.000	0.000		0.16
PWQ	0.000	0.000	0.000	0.000	0.004	0.000	0.000	0.000	0.000	
**(b)** **Correlation table for precipitation, temperature-related variables, calculated indices, and elevation.**										
**Variable**	**AMT**		**MaxT**	**HITmead**	**HITmax**	**AP**	**DI**		**ELEV**	
AMT			0.98	0.92	0.93	0.07	0.42		−0.16	
MaxT	0.000			0.92	0.96	−0.02	0.49		−0.14	
HITmead	0.000		0.000		0.97	0.06	0.45		−0.38	
HITmax	0.000		0.000	0.000		−0.05	0.53		−0.27	
AP	0.023		0.602	0.042	0.117		−0.83		−0.13	
DI	0.000		0.000	0.000	0.000	0.000			−0.11	
ELEV	0.000		0.000	0.000	0.000	0.000	0.000			

**Table 2 plants-11-02289-t002:** Number of genes detected from Genome–environment Associations (GEA) for seven climatic variables with 26,064 SNPs in 990 landraces of bread wheat.

Chr	Physical Position (bp)	SNP	Variables	Genes
1A	277825486	108825112|F|0-19:T>A-19:T>A	AMT, MaxT y HITmax	1
1A	388440316	108891114|F|0-33:A>G-33:A>G	AMT, MaxT, HITmead y HITmax	1
1A	588647447	108256081|F|0-24:C>G-24:C>G	AP	1
1B	201700273	108853213|F|0-26:C>T-26:C>T	HITmead	1
1D	2113561	108739422|F|0-31:T>A-31:T>A	AMT	1
1D	413939387	107880401|F|0-66:C>T-66:C>T	HITmead	2
1D	485539732	107874524|F|0-37:G>C-37:G>C	PS	3
2A	640221452	109058266|F|0-7:C>T-7:C>T	AP	2
2A	711971015	108020469|F|0-43:T>C-43:T>C	PS	2
2A	729345577	108514047|F|0-10:A>C-10:A>C	PS	2
2B	29129965	107488994|F|0-17:T>C-17:T>C	HITmead	3
2B	38321752	108024073|F|0-47:C>T-47:C>T	HITmax	2
2B	76022529	107794074|F|0-23:T>C-23:T>C	PS	2
2B	91596926	107797386|F|0-11:T>G-11:T>G	AP	1
2B	584134131	108980638|F|0-33:T>C-33:T>C	DI	1
2B	768567114	107593284|F|0-50:T>A-50:T>A	AP	1
2B	795754781	109021888|F|0-34:T>C-34:T>C	PS	1
2D	79989524	108968402|F|0-23:C>T-23:C>T	AMT	2
2D	302776397	107489027|F|0-34:C>T-34:C>T	DI	1
3A	502971897	106510612|F|0-30:C>T-30:C>T	MaxT	2
3A	507114154	108476623|F|0-32:T>A-32:T>A	MaxT y HITmax	2
3A	574516154	108028914|F|0-20:A>G-20:A>G	MaxT y HITmax	2
3B	50526285	108953425|F|0-21:A>G-21:A>G	PS	1
3B	535641207	109122135|F|0-40:A>G-40:A>G	HITmax	1
3B	741467423	107601308|F|0-23:G>A-23:G>A	PS	0
3B	758205945	108516380|F|0-24:T>C-24:T>C	PS	1
3B	764282419	108146856|F|0-65:T>A-65:T>A	HITmead	1
3D	97153088	107698139|F|0-48:C>T-48:C>T	AMT y HITmead	1
3D	546672913	108308207|F|0-18:G>A-18:G>A	AMT y HITmax	2
3D	575639014	109610030|F|0-20:C>T-20:C>T	HITmax	2
4A	598521397	109242168|F|0-13:A>G-13:A>G	HITmead y HITmax	2
4A	714179675	108145069|F|0-52:C>A-52:C>A	AP	2
4B	39759168	106772473|F|0-11:T>G-11:T>G	PS	1
4B	56276689	108145974|F|0-24:T>C-24:T>C	HITmead	1
4D	62303691	109363420|F|0-34:G>A-34:G>A	HITmead	2
4D	97959582	108773963|F|0-37:C>T-37:C>T	HITmead	2
4D	370691683	109431634|F|0-54:G>A-54:G>A	AMT, MaxT, HITmead y HITmax	3
5A	619468262	107001591|F|0-37:T>C-37:T>C	PS	2
5B	68925700	109240982|F|0-38:G>C-38:G>C	HITmead	1
5B	526249013	109305833|F|0-34:C>G-34:C>G	AP y DI	1
5B	530916378	108773279|F|0-51:C>T-51:C>T	HITmead	2
5B	548120559	108348543|F|0-58:A>G-58:A>G	PS	1
5B	577227418	109119258|F|0-68:T>C-68:T>C	PS	2
5D	379130055	108305241|F|0-68:A>G-68:A>G	AMT	1
5D	528660566	108652995|F|0-58:G>C-58:G>C	MaxT	1
6A	5854616	109126792|F|0-49:C>T-49:C>T	MaxT	2
6B	223441723	109177937|F|0-11:A>G-11:A>G	AP	1
6B	269818400	109523315|F|0-24:C>T-24:C>T	AP	0
6B	485290761	109354013|F|0-58:A>G-58:A>G	PS	2
6D	143959936	108582786|F|0-49:T>G-49:T>G	MaxT y HITmax	2
7A	64789408	108830300|F|0-9:T>G-9:T>G	MaxT	2
7A	67617920	108206445|F|0-49:C>A-49:C>A	DI	1
7A	498861613	107952026|F|0-63:G>C-63:G>C	DI	1
7A	552606647	108619258|F|0-17:C>T-17:C>T	PS	1
7A	628906923	107178221|F|0-24:T>G-24:T>G	HITmead	2
7A	662017143	109126469|F|0-35:G>A-35:G>A	MaxT	2
7B	107522176	107878167|F|0-16:G>A-16:G>A	MaxT	2
7B	650581291	108981313|F|0-8:C>T-8:C>T	PS	1
7D	88318125	109035950|F|0-29:C>T-29:C>T	DI	1
			**Total**	**89**

**Table 3 plants-11-02289-t003:** Signaling genes and proteins identified for Genome–environment Associations (GEA) with seven climatic variables.

Chr	SNP	Gene	Variable	Protein	Function
1A	108825112|F|0-19:T>A-19:T>A	TraesCS1A02G156600	AMT, MaxT and HITmax	Auxin response factor (*ARF*)	Auxin-activated signaling.
1D	108739422|F|0-31:T>A-31:T>A	TraesCS1D02G003900	AMT	Peroxisome membrane anchor (*PEX14*).	Transduction of stress signals by *ROS*.
1D	107880401|F|0-66:C>T-66:C>T	TraesCS1D02G319600	HITmead	S-adenosyl-L-methionine-dependent methyltransferases (*SAMe*).	Methylation of DNA and proteins, *ET* biosynthesis, phenylpropanoid biosynthesis.
1D	107874524|F|0-37:G>C-37:G>C	TraesCS1D02G438700	PS	Swi-Independent 3 (SIN3)-Like 1 (*SNL1*).	*ABA* and *ET*-activated signaling.
2A	108020469|F|0-43:T>C-43:T>C	TraesCS2A02G467400	PS	Protein enhanced pseudomonas susceptibility 1 (*EPS1*).	*SA* biosynthesis and response to *JA*.
2A	108514047|F|0-10:A>C-10:A>C	TraesCS2A02G500200	PS	Ser/Thr phosphatase (*PstP*).	Signaling in response to *ABA*.
2B	108024073|F|0-47:C>T-47:C>T	TraesCS2B02G071900	HITmax	Ser/Thr kinase (*STPK*).	Signaling cascades.
2B	107794074|F|0-23:T>C-23:T>C	TraesCS2B02G112600	PS	MYB108 TF (*MYB*).	Response to signaling by *ABA* and *JA*.
2B	107797386|F|0-11:T>G-11:T>G	TraesCS2B02G123900	PS	Nonspecific Ser/Thr kinase (*CIPK2*).	Signaling cascades.
2D	107489027|F|0-34:C>T-34:C>T	TraesCS2D02G252400	DI	Sugar/inositol transporter 2 (*INT2*).	Transduction of hormonal signals.
3A	106510612|F|0-30:C>T-30:C>T	TraesCS3A02G274000	MaxT	Ser/Thr kinase (*STPK*).	Signaling cascades.
3B	109122135|F|0-40:A>G-40:A>G	TraesCS3B02G331800	HITmax	Pentatricopeptide repeat (*PPR*).	Signaling in response to *ABA*.
3B	108516380|F|0-24:T>C-24:T>C	TraesCS3B02G516800	PS	Mitochondrial ribosomal S4 (*RPS4*).	Proteins encoded in the mitochondrial genome exported to the cytoplasm.
3D	108308207|F|0-18:G>A-18:G>A	TraesCS3D02G433200	AMT and HITmax	Similar to helix-loop-helix DNA (*bHLH*)	Signaling in response to auxin and cell wall modification.
4A	109242168|F|0-13:A>G-13:A>G	TraesCS4A02G301600 TraesCS4A02G302000	HITmead and HITmax	2-methyl-6-phytyl-1,4-hydroquinone methyltransferase (*VTE3*). Ser/Thr kinase (*STPK*).	Vitamin *E* biosynthesis and stress signaling. Signaling cascades.
4D	109363420|F|0-34:G>A-34:G>A	TraesCS4D02G087000	HITmead	Ethylene-responsive TF (*ERF022*).	*ET*-activated signaling.
4D	109431634|F|0-54:G>A-54:G>A	TraesCS4D02G216300 TraesCS4D02G216600	AMT, MaxT, HITmead and HITmax	Ethylene-responsive TF (*ERF014*). Ser/Thr kinase (*STPK*).	*ET*-activated signaling. Signaling cascades.
5B	108348543|F|0-58:A>G-58:A>G	TraesCS5B02G369300	PS	Metallophone domain (*MPE*).	GPI biosynthesis the cell membrane.
7A	107952026|F|0-63:G>C-63:G>C	TraesCS7A02G340300	DI	Basic helix-loop-helix (*bHLH*).	Signaling in response to auxin and cell wall modification.
7A	108619258|F|0-17:C>T-17:C>T	TraesCS7A02G377500	PS	EIN3-binding F-box 1 (*EBF1*).	ET-activated signaling.
7A	107178221|F|0-24:T>G-24:T>G	TraesCS7A02G435700	HITmead	IAA-amino acid hydrolase (*ILL*).	Auxin metabolic process.
7A	109126469|F|0-35:G>A-35:G>A	TraesCS7A02G465400 TraesCS7A02G465500	MaxT	Kinase (Kinase). Glycine-rich domain 2 (*GRDP2*).	Signaling cascades. Auxin-activated signaling.
7B	108981313|F|0-8:C>T-8:C>T	TraesCS7B02G385700	PS	HTH myb-type domain (*MYB*).	Response to signaling by *ABA* and *JA*.

**Table 4 plants-11-02289-t004:** Cell wall genes and proteins identified for Genome–environment Associations (GEA) with seven climatic variables.

Chr	SNP	Gene	Variable	Protein	Function
1A	108256081|F|0-24:C>G-24:C>G	TraesCS1A02G439300	AP	ArfGAP domain 2G (*ArfGAP*).	Membrane trafficking.
1D	107874524|F|0-37:G>C-37:G>C	TraesCS1D02G439900	PS	RING-CH-type domain/E3 ubiquitin ligase (*MARCH*).	Protein degradation by the ubiquitin pathway with abnormalities.
2B	107488994|F|0-17:T>C-17:T>C	TraesCS2B02G059400	HITmead	Wall-associated kinase (*WAK*).	Regulation of wall functions and signaling of extracellular environment.
2B	108024073|F|0-47:C>T-47:C>T	TraesCS2B02G071600	HITmax	Acyl-CoA--sterol O-acyltransferase (*ASAT1*).	Synthesis of long-chain esters (waxes).
2B	109021888|F|0-34:T>C-34:T>C	TraesCS2B02G621600	PS	C2 calcium/lipid-phosphoribosyltransferase (*QKY*).	Signal transduction or calcium-dependent membrane trafficking.
2D	108968402|F|0-23:C>T-23:C>T	TraesCS2D02G136300	AMT	Glycosyltransferase STELLO2 (*STL2*).	Cell wall cellulose biosynthesis.
3A	106510612|F|0-30:C>T-30:C>T	TraesCS3A02G274200	MaxT	Mannan endo-1,4-beta-mannosidase 2 (*MAN2*).	Lignocellulose component in primary cell walls.
3A	108476623|F|0-32:T>A-32:T>A	TraesCS3A02G277100	MaxT and HITmax	Glucan endo-1,3-beta-glucosidase 13 (*1,3-**β**-glucanasa*).	Defense against pathogens, cell wall biogenesis and reorganization.
3D	107698139|F|0-48:C>T-48:C>T	TraesCS3D02G138700	AMT and HITmead	O-methyltransferase (*OMT*).	Lignin biosynthesis.
3D	108308207|F|0-18:G>A-18:G>A	TraesCS3D02G433400	AMT and HITmax	Glycosyltransferase (*GT*).	Biosynthesis polysaccharides of cell walls: cellulose, hemicellulose, and pectin.
3D	109610030|F|0-20:C>T-20:C>T	TraesCS3D02G474800	HITmax	Putative expansin-B2 (*EXPB2*).	Loosening of plant cell walls.
5B	109240982|F|0-38:G>C-38:G>C	TraesCS5B02G061500	HITmead	Root hair defective 3 (*RDH2*).	Biogenesis of the cell wall and organization of the cytoskeleton.
5D	108305241|F|0-68:A>G-68:A>G	TraesCS5D02G276400	AMT	Cinnamoyl-CoA reductase 4 (*CCR*).	Primary alcohols and leaf cuticular wax synthesis.
7A	108830300|F|0-9:T>G-9:T>G	TraesCS7A02G107500	MaxT	3-ketoacyl-CoA synthase (*KCS*).	Synthesis of long-chain esters (waxes).
7B	107878167|F|0-16:G>A-16:G>A	TraesCS7B02G093900	MaxT	Glyco_trans_2-like (*GT2*).	Cell wall organization.

**Table 5 plants-11-02289-t005:** Abiotic stress genes and proteins identified for Genome–environment Associations (GEA) with seven climatic variables.

Chr	SNP	Gene	Variable	Protein	Function
1B	108853213|F|0-26:C>T-26:C>T	TraesCS1B02G146100	HITmead	40S ribosomal S7 (*RPS7*).	Response to environmental signals.
1D	107880401|F|0-66:C>T-66:C>T	TraesCS1D02G319400	HITmead	Heat shock protein class VI (*HSP20*).	Heat and salt tolerance.
1D	107874524|F|0-37:G>C-37:G>C	TraesCS1D02G439800	PS	Trimethylguanosine synthase (*TGS*).	Cold tolerance.
2B	107794074|F|0-23:T>C-23:T>C	TraesCS2B02G112800	PS	Galactinol synthase 7 (*GolS*).	Tolerance to drought, salinity, and cold.
2B	107593284|F|0-50:T>A-50:T>A	TraesCS2B02G581100	AP	Leucine-Rich Repeat Kinase (*LRRK1*).	Tolerance to drought.
3A	108476623|F|0-32:T>A-32:T>A	TraesCS3A02G276800	MaxT and HITmax	Glutamate carboxypeptidase (*AMP1*).	Responses to oxidative stress.
4A	108145069|F|0-52:C>A-52:C>A	TraesCS4A02G446900	AP	Cytochrome P450 709B3 (*CYP450*).	Biosynthesis of secondary metabolites and phytohormones in response to stress.
4D	108773963|F|0-37:C>T-37:C>T	TraesCS4D02G117200	HITmead	Peroxidase (*POD*).	Response to oxidative stress.
4D	109431634|F|0-54:G>A-54:G>A	TraesCS4D02G216700	AMT, MaxT, HITmead and HITmax	Late embryogenesis abundant (*LEA*).	Osmotic stress.
5A	107001591|F|0-37:T>C-37:T>C	TraesCS5A02G437900	PS	Heat shock factor (*HSF*).	Heat shock proteins.
5B	109305833|F|0-34:C>G-34:C>G	TraesCS5B02G341100	AP and DI	GrpE protein homolog (*GrpE*).	Thermotolerance to chronic heat stress.
5B	108773279|F|0-51:C>T-51:C>T	TraesCS5B02G350000	HITmead	BURP domain (*BURP*).	Responses to drought stress by *ABA*.
5B	109119258|F|0-68:T>C-68:T>C	TraesCS5B02G399900	PS	Probable WRKY TF 57 (*WRKY*).	Response to osmotic stress, salt, and drought.
5D	108652995|F|0-58:G>C-58:G>C	TraesCS5D02G498900	MaxT	DNA-directed RNA polymerase subunit (*NRPB1*).	Response to heat stress.
6A	108582786|F|0-49:T>G-49:T>G	TraesCS6A02G012100 TraesCS6A02G013100	MaxT	Cytochrome P450 709B1 (*CYP450*). Leucine-rich repeat receptor-like kinase (*LRRK2*).	Biosynthesis of secondary metabolites and phytohormones in response to stress. Tolerance to drought.
6B	109354013|F|0-58:A>G-58:A>G	TraesCS6B02G269500	PS	Cytochrome P450 (*CYP450*).	Biosynthesis of secondary metabolites and phytohormones in response to stress.
6D	108582786|F|0-49:T>G-49:T>G	TraesCS6D02G164900 TraesCS6D02G165100	MaxT and HITmax	Adenine nucleotide alpha hydrolases (*ANAH*). Cytochrome P450 (*CYP450*).	Response to salt stress. Biosynthesis of secondary metabolites and phytohormones in response to stress.
7A	108206445|F|0-49:C>A-49:C>A	TraesCS7A02G110500	DI	F-box component of the SKP-Cullin-F-box E3 (*SCF*).	Water deficit.
7A	107178221|F|0-24:T>G-24:T>G	TraesCS7A02G435300	HITmead	Leucine-rich repeat receptor-like kinase (*LRRK3*).	Tolerance to drought.

**Table 6 plants-11-02289-t006:** Landrace countries of origin and geographic regions.

Country Code	Country	Region	Region Code	Total Landraces
AFG	Afghanistan	Southern Asia	SAS	51
ARM	Armenia	Western Asia	WAS	7
AUS	Australia	Australia and New Zealand	AUS	1
AUT	Austria	Western Europe	WEU	1
AZE	Azerbaijan	Western Asia	WAS	5
CAN	Canada	Northern America	NAM	1
CHN	China	Eastern Asia	EAS	275
DEU	Germany	Western Europe	WEU	1
DZA	Algeria	Northern Africa	NAF	1
ESP	Spain	Southern Europe	SEU	3
ETH	Ethiopia	Eastern Africa	EAF	4
GEO	Georgia	Western Asia	WAS	10
GRC	Greece	Southern Europe	SEU	1
IND	India	Southern Asia	SAS	17
IRN	Iran	Southern Asia	SAS	34
ESP	Spain	Southern Europe	SEU	3
ETH	Ethiopia	Eastern Africa	EAF	4
GEO	Georgia	Western Asia	WAS	10
GRC	Greece	Southern Europe	SEU	1
IND	India	Southern Asia	SAS	17
IRN	Iran	Southern Asia	SAS	34
IRQ	Iraq	Western Asia	WAS	8
ITA	Italy	Southern Europe	SEU	1
JPN	Japan	Eastern Asia	EAS	1
LBN	Lebanon	Western Asia	WAS	1
MEX	Mexico	Central America	CAM	9
PAK	Pakistan	Southern Asia	SAS	1
PER	Peru	South America	SAM	1
POL	Poland	Eastern Europe	EEU	1
PRT	Portugal	Southern Europe	SEU	8
RUS	Russia	Eastern Europe	EEU	2
SRB	Serbia	Southern Europe	SEU	1
SYR	Syria	Western Asia	WAS	3
TJK	Tajikistan	Central Asia	CAS	42
TUN	Tunisia	Northern Africa	NAF	2
TUR	Turkey	Western Asia	WAS	490
USA	United States	Northern America	NAM	4
UZB	Uzbekistan	Central Asia	CAS	2
VEN	Venezuela	South America	SAM	1

**Table 7 plants-11-02289-t007:** List of geographic, climatic, and bioclimatic variables downloaded from the WorldClim platform.

Abbreviation	Variable Description and Unit
ELEV	Altitude, meters.
MaxT	Maximum temperature, °C × 10.
AMT	Annual mean temperature, °C × 10.
TS	Temperature seasonality, standard deviation × 100.
MaxTWM	Maximum temperature of warmest month, °C × 10.
MeanTDQ	Mean temperature of driest quarter, °C × 10.
MeanTWQ	Mean temperature of warmest quarter, °C × 10.
AP	Annual precipitation, mm.
PDM	Precipitation of driest month, mm.
PS	Precipitation seasonality, mm.
PDQ	Precipitation of driest quarter, mm.
PWQ	Precipitation of warmest quarter, mm.

**Table 8 plants-11-02289-t008:** Climate types under the Köppen-Geiger climate classification.

Group	Climates
A—Tropical	Tropical rainforest (*Af*), Tropical monsoon (*Am*), and Tropical savanna (*Aw*, *As*).
B—Dry	Desert (*BWh*, *BWk*) and Semi-arid (*BSh*, *BSk*).
C—Temperate	Humid subtropical (*Cfa*, *Cwa*); Oceanic (*Cfb*, *Cwb*, *Cfc*, *Cwc*); and Mediterranean (*Csa*, *Csb*, *Csc*).
D—Continental	Humid continental (*Dfa*, *Dwa*, *Dfb*, *Dwb*, *Dsa*, *Dsb*) and Subarctic (*Dfc*, *Dwc*, *Dfd*, *Dwd*, *Dsc*, *Dsd*).
E—Polar	Tundra (*ET*), Ice cap (*EF*), and Alpine (*ET*, *EF*).

## Data Availability

The data presented in this study are available on request from the corresponding author.

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
