# Peer review of "Worldwide Selection Footprints for Drought and Heat in Bread Wheat (Triticum aestivum L.)"

_plants, 2022, doi:10.3390/plants11172289_

Round 1

Reviewer 1 Report

This is a well-written manuscript exploring the genetic basis of adaptive traits in bread wheat.  In this work, two powerful modern genomic strategies such as GWAS (genome-wide association study) and GEA (genome-environment association) approaches were coupled in a study of a diverse set of 990 wheat landraces to identify associations between SNP alleles and the environment of origin in the landraces originating from tropical to polar environments. Significant SNP markers associated with environmental variables were identified. A number of putative candidate genes involved in plant adaptation responses to abiotic stress were suggested including genes related to drought and heat stress tolerance. For crop improvement, it is of supreme importance to identify loci associated with adaptive phenotypes in staple crops, particularly at the face of the current climate change and the urgent need to feed the expanding human population. The applied GEAs approach has potential to predict factors associated with plant adaptation to abiotic stress. In this regard, my opinion is that the manuscript merits publication.

The Introduction is concise but well informative. The aims are defined logically and clearly. The plant material and applied methods are adequate. The results are sufficient and well-presented. The discussion is rather short, could be extended. The Abstract reflects the research problem, the major findings and their perspectives.

I have only a few minor remarks:

1)      All figures and tables should be self-explanatory. I suggest that authors either define the used abbreviations in the captions, or to refer to section Material and Methods.

2)      Line 91: replace “y” to “and”

3)      Line 114: Table 6 is referred to before Tables 2 to 5. Please, correct the numbering of Tables throughout the text.

4)      Table 2: Please, explain why some SNPs are given in bold. Also, in the caption of the same table, the “in 990 collections of bread wheat” is not correct. Maybe, in a collection of 990 landraces?

5)      Captions of Figures 4 and 5: “990 flour collections”. What does it mean? Is it correct?

6)      Caption of Figure 6: “heat and drought genes” is not correct. Please, modify. Maybe, “heat and drought tolerance related” genes? Also, in the line below (line 246): “de SNP”. Please, correct.

7)      Line 258: “structure of wild accessions” change to “population structure of wild accessions”

8)      Line 305: modify to “such as…..”

9)      Lines 337-338: Two identical sentences, but different number of accessions are mentioned. Please correct.

10)   Lines 353-354 contain unclear/unfinished sentence: “The filtered table contained 26, 064 SNP loci, from which a subset containing only accessions was selected with geographic data”. Please, check and correct.

Author Response

We attended the list of observations and made the corrections.

1) All figures and tables should be self-explanatory. I suggest that authors either define the used abbreviations in the captions, or to refer to section Material and Methods.

R. In this version we include the definition of the acronym GEA in the captions of tables and figures.

2) Line 91: replace “y” to “and”

R. In line 91, the word "y" was replaced by "and".

3) Line 114: Table 6 is referred to before Tables 2 to 5. Please, correct the numbering of Tables throughout the text.

R. In line 114, It was unnecessary to mention "Table 6" in the text, , so it was eliminated.

4) Table 2: Please, explain why some SNPs are given in bold. Also, in the caption of the same table, the “in 990 collections of bread wheat” is not correct. Maybe, in a collection of 990 landraces?

R. The most frequent SNPs, mentioned in lines 168 and 169, were highlighted in bold in Table 2. However, to avoid confusion, the highlighting was omitted. In the header of the same table (line 173), the word "collections" was replaced by "landraces".

5) Captions of Figures 4 and 5: “990 flour collections”. What does it mean? Is it correct?

R. In captions of Figures 4 and 5, the phrase "990 flour collections" was replaced by "landraces of bread wheat" (line 181).

6) Caption of Figure 6: “heat and drought genes” is not correct. Please, modify. Maybe, “heat and drought tolerance related” genes? Also, in the line below (line 246): “de SNP”. Please, correct.

R. Caption of Figure 6, the phrase "heat and drought genes" was changed to "heat and drought tolerance-related genes" (line 256). In line 257, the word "de" removed in the same figure.

7) Line 258: “structure of wild accessions” change to “population structure of wild accessions”

R. The phrase "structure of wild accessions" in line 268 was changed to "population structure of wild accessions".

8) Line 305: modify to “such as…..”

R. In line 317, the word "such…" was changed to " such as…".

9) Lines 337-338: Two identical sentences, but different number of accessions are mentioned. Please correct.

R. In line 355, the phrase "This filtering process yielded 1,151 landraces with validated geographic data" was removed from the text; on the contrary, the expression "1,151 landraces of …" was added in line 352.

10) Lines 353-354 contain unclear/unfinished sentence: “The filtered table contained 26, 064 SNP loci, from which a subset containing only accessions was selected with geographic data”. Please, check and correct.

R. In line 370, the table filtering was performed for the 990 landraces with geo-referenced data. We have clarified this point.

Reviewer 2 Report

The authors identified 59 SNPs, and near 89 protein-encoding genes involved in response processes to abiotic stress. In addition, these genes related to biosynthesis and signaling are mainly mediated by auxins, abscisic acid, ethylene, salicylic acid, and jasmonates, which are known to operate together in modulation responses to heat stress and drought in plants. In addition, we identified some proteins associated with the response and tolerance to stress by high temperatures, water deficit, and cell wall functions. In general, this work is interesting, and provides further understanding on wheat response to drought and heat stresses. I have two minor concerns that need to be addressed.

1. The authors should select some genes for RT-qPCR to verify whether these candidate genes have obvious changes under drought and heat stresses.

2. Some figures are too small to see clearly and need to be enlarged.

Author Response

We are in the process of analyzing the candidate genes. Some of them may imply quantitative changes in expression, susceptible to be checked by RT-qPCR. Others may lead to protein sequence variation. This is, however, a goal for a further investigation in this project.

You are right, the figures had some problems, mainly with font size. In this version we are providing figures with higher resolution and larger font size.

Reviewer 3 Report

This manuscript reports results of association between sequence polymorphims and climatic variables of their site of origin, usin 990 wheat landraces collected worldwide. Personnaly,I would not have called that "selection footprint", but rather genetic association, as selection footprints can be detected from the study of polymorphism alone, whiththout explanatory variable. But if this is now admitted, I will accept too.

First of all, I must say that I dislike article where Materials and methods are at the end. This led to continuous go and back, for example to understand labels, to unlogical numbering of table, e.g. Table 6 cited l 114, before Table 2! while it apperas only in MM; But authors are not responsible

Another drwback is that we have to wait until MM to understand concepts which are cited before, but not describe, or even sigle which are not developped, enven in the abstract (GAPIT, FarmCPU...)

To complete my detail remarks, I was not able to read Figure 3b: symbols are really TOO small. And some citations seem to be strange. For example for QGIS, I was able to see the website (thank to ggogle), but no mention of "Las Palmas"

To come to my main remark: in Table 6 listing the origin of landraces, there is a clear over-representation of Turkey and China. Turkey may have some logics, as it hosts the craddle of wheat domestication. And Chine, beside being an important center of secondary diversification, is likely to have cultivated landraces later than Europe.

But my main question is about the time period considered for possible selection. The authors considered 1970-2000 for climatic variables or indices. It seems to be a rather short period, considering the time landraces have experienced selection (at least one century in Australia, millenaries in Europe/Asia). But more importantly, nothig is said about the time lanraces wre collected. During their storage in CIMMYT genebank, one can postulate their variability was "frozen". Therefore, the 30 (or more) years BEFORE collection time should be considered, instead of a fixed period for every landrace

Overall, the authors have identified 89 genes with significant association with climate. This may have an intereset for breeders, for example to introduce heat-adapted alleles into current temperate germplasm.I would have appreciated to see the geographic variation illustrated on a map, for one or two most significant associations.

Provided the authors adress my comments and main question about collecting time, I consider this manuscript should be worthwhile to be published

Author Response

We agree that placing the materials and methods at the end of the document may create confusion. However, we followed the MDPI guidelines.

The figures' resolution, and font size have been increased to improve their visualization.

The reason for the data collection period was because the historical records of high-resolution bioclimatic variables stored in the WordClim database only cover 1970-2000. For this reason, most Genome-environment associations (GEA) reported in the literature have been handled with climatic data from this period. This source of data has been useful for identifying SNP markers associated with traits of adaptive importance. It would have been interesting to use a more extended period to collect climatic data, however we think that conditions during those 30 years at least indicate accurately the type of climate prevailing in the studied regions.

We added a paragraph about the representativeness of the 30-year period of climate records. We also provide information about collection dates, and the drawbacks resulting from the dynamics of wheat populations.

It should be remarked that climate data come from records spanning 1970-2000, which naturally do not cover the adaptation period of the accessions. However, these 30-year records are good indicators of the prevailing climate type in the collection sites. On the other hand, the accessions were collected from 1983 to 2011, with the main bulk of collection occurring in 1984 and 2011. One must be aware of the noise arising from the migration dynamics of those materials, especially for the most recent collection efforts. Thus, the herein reported results are subject to validation by different approaches.”

Round 2

Reviewer 3 Report

The authors did their best to follow my recommendations. I was aware this was not always possible, in particular for historical climatic data.

I consider this manuscript is now worthwhile to be published inits present form